# Integrated VIS/NIR Spectrum and Genome-Wide Association Study for Genetic Dissection of Cellulose Crystallinity in Wheat Stems

**DOI:** 10.3390/ijms25053028

**Published:** 2024-03-06

**Authors:** Jianguo Li, Peimin Zhao, Liyan Zhao, Qiang Chen, Shikun Nong, Qiang Li, Lingqiang Wang

**Affiliations:** 1College of Plant Science & Technology, Huazhong Agricultural University, Wuhan 430070, China; 15611821765@163.com (J.L.); zhaopeimin@webmail.hzau.edu.cn (P.Z.); liyanzhau@163.com (L.Z.); 2State Key Laboratory for Conservation & Utilization of Subtropical Agro-Bioresources, College of Agriculture, Guangxi University, Nanning 530004, China; 19114787704@163.com (Q.C.); 15278698778@163.com (S.N.)

**Keywords:** cellulose crystallinity, wheat, VIS/NIR, GWAS

## Abstract

Cellulose crystallinity is a crucial factor influencing stem strength and, consequently, wheat lodging. However, the genetic dissection of cellulose crystallinity is less reported due to the difficulty of its measurement. In this study, VIS/NIR spectra and cellulose crystallinity were measured for a wheat accession panel with diverse genetic backgrounds. We developed a reliable VIS/NIR model for cellulose crystallinity with a high determination coefficient (R^2^) (0.95) and residual prediction deviation (RPD) (4.04), enabling the rapid screening of wheat samples. A GWAS of the cellulose crystallinity in 326 wheat accessions revealed 14 significant SNPs and 13 QTLs. Two candidate genes, *TraesCS4B03G0029800* and *TraesCS5B03G1085500,* were identified. In summary, this study establishes an efficient method for the measurement of cellulose crystallinity in wheat stems and provides a genetic basis for enhancing lodging resistance in wheat.

## 1. Introduction

Wheat makes a substantial contribution to agriculture’s sustainability and plays a major role in ensuring food/nutrition security worldwide [1]. Remarkably, China is considered the largest wheat producer and consumer globally, and lodging is still a fundamental challenge in wheat cultivation [2]. Lodging, such as stem breaking, severely decreases the quality and quantity of wheat yield [3]. In wheat, traits associated with lodging resistance could be grouped into morphological, anatomical, and cell wall component levels [4]. Among these three levels, the genetic dissection of cell wall components, especially fine structures like cellulose crystallinity, has been less explored due to the difficulty in its measurement.

The plant cell wall is primarily composed of cellulose, hemicellulose, and lignin. The arrangement and interaction of the components in the culm cell wall are of great importance for lodging resistance in wheat [4]. Cellulose forms the main structural framework of the cell wall, and cellulose crystallinity, representing the ratio of crystalline to non-crystalline regions, determines cellulosic fibril properties such as tensile strength [5]. Cellulose crystallinity is the major factor that negatively determines breaking-type lodging resistance [6]. Interestingly, brittle culm mutants of rice have been shown to significantly reduce cellulose crystallinity, and many genes have been cloned, such as *OsBC1* (COBRA-like protein), *OsBC3* (classical dynamin protein family), and *OsBC6* (cellulose synthase) [7,8,9]. However, few genes are available for cellulose crystallinity in wheat.

Recently, genome-wide association studies (GWASs) have become a powerful tool for assessing complex traits [10,11]. Although GWASs face problems in explaining environmental effects, missing heritability, detecting false positives of significant loci, expanding the population size, increasing the number of markers, increasing the threshold, and collecting accurate phenotypes in different environments could make GWAS results more reliable. The release of the high-quality Chinese spring reference genome [12], provides the availability of large-scale sequencing data for wheat [8], which contributes to the application of GWASs in wheat. By now, GWASs have been widely used in wheat [10,13,14]. Furthermore, the high-throughput phenotyping method greatly broadens the usage of GWASs in wheat [10]. Thus, GWASs could be an effective method for the genetic dissection of cellulose crystallinity, but the traditional method for measuring cellulose crystallinity is expensive and time-consuming [15]. 

The traditional methods for determining complex traits such as cell wall components are labor-intensive and expensive, which is not suitable for large-scale studies [11]. Near-infrared spectra (NIR), combined with spectra data and laboratory data, are well suited for carrying out high-efficiency chemical detection [16]. Studies have shown that NIR or VIS/NIR models can be used for the determination of cell wall components such as cellulose and lignin [17,18,19]. Nevertheless, most NIR or VIS/NIR models were used for the prediction of cell wall components, while models for cellulose crystallinity have been reported less [20,21]. Further, little is reported about NIR models for cellulose crystallinity in wheat, especially on a large scale. Therefore, methods for the rapid measurement of cellulose crystallinity are still needed for wheat. 

Crystalline cellulose affects the strength of wheat stems. However, little has been reported on cellulose crystallinity in wheat. Here, we collected the spectra of 326 wheat accessions and measured the cellulose crystallinity index (CrI) of 145 wheat accessions. Then, NIR models were built and evaluated for cellulose crystallinity, and the best model was used for the cellulose crystallinity prediction of the remaining wheat accessions. Furthermore, a GWAS was carried out on cellulose crystallinity across the whole association panel. Hence, we established an effective method for the large-scale screening of cellulose crystallinity and provided a genetic basis for lodging resistance breeding in wheat. Our results demonstrate that combining NIR spectra and a GWAS provides new insight into the genetic basis of cellulose crystallinity in wheat.

## 2. Results

### 2.1. Diversity, Prediction, and Description of Cellulose CrI

The association panel consisted of a diverse collection of 326 wheat accessions (including 268 cultivars, 57 landraces, and an unclear type accession) (Appendix A). The cellulose crystallinity was measured in 145 wheat accessions by an X-ray diffraction method (Figure 1A). The cellulose CrI of the 145 accessions ranged from 0.32 to 0.54, and the mean value of cellulose CrI was 0.46, which is similar to barley (0.43) [21]. The coefficient of variation was 8.17%, and the absolute values of skewness (−0.85) and kurtosis (1.13) were smaller than 2, suggesting that cellulose CrI is normally distributed. Therefore, cellulose CrI had a relatively normal distribution and a great variation, which were applied to the VIS/NIR model. 

The spectra of 326 wheat accessions were collected, and baseline correction was carried out on the raw spectra. Furthermore, pretreatments (standard normal variation (SNV), multiple scatter correction (MSC), first derivative (FD), and Savitzky–Golay filtering (SG)) were performed, and the transformed spectra showed a more diverse variation than the raw spectra (Appendix A). Previous studies found that a support vector machine regression (SVR) model is suitable for the prediction of a relatively small sample size, and SVR has been widely used in VIS/NIR models [22,23]. In this study, four models showed high determination coefficients (R^2^) (>0.95) in the calibration dataset. In detail, the determination coefficient (R^2^) ranged from 0.95 to 0.99, and the first derivative (FD) treatment had the highest determination coefficient (R^2^) (0.99), while the Savitzky–Golay (SG) pretreatment had the smallest determination coefficient (R^2^) (0.95). The residual prediction deviation (RPD) ranged from 4.09 to 10.90, and the first derivative (FD) treatment had the highest RPD (10.90), while the Savitzky–Golay (SG) method had the smallest RPD (4.09). Moreover, the root mean square error (RMSE) of the four pretreatments in the calibration dataset was smaller than 0.01. As for the validation dataset, the determination coefficient (R^2^) ranged from 0.85 to 0.95, and the multiple scatter correction (MSC) pretreatment showed the highest determination coefficient (R^2^) (0.95), while the Savitzky–Golay (SG) method had the smallest determination coefficient (R^2^) (0.85). The residual prediction deviation (RPD) ranged from 2.63 to 4.04, and the multiple scatter correction (MSC) pretreatment also had the highest RPD (4.04), while the Savitzky–Golay (SG) filter had the smallest RPD (2.63). The root mean square error (RMSE) of the four pretreatments in the validation process was smaller than 0.02. (Figure 1B, Appendix A). Thus, the MSC method was used for the cellulose CrI prediction of the association panel.

The cellulose CrI of the whole association panel followed a normal distribution (Figure 1C), and the cellulose CrIof the whole association panel also had great coefficient variation, which was great for the GWAS (Appendix A). The cellulose CrI of the association panel ranged from 0.32 to 0.59, and ADDIS ABEDA 26 (foreign landrace accession) had the highest cellulose CrI (0.59), while Jinmai-31 (a Chinese cultivar) had the smallest cellulose CrI (0.32). The coefficient of variation of the whole association panel was 7.42%, while the skewness and kurtosis were −0.3558 and 1.297, respectively. Moreover, the landrace accessions had a higher coefficient variation (8.26%) than the cultivar accessions (7.22%), while the range of cultivar accessions (0.32–0.57) was larger than that of the landrace accessions (0.37–0.59). The cellulose CrI results showed no difference between the cultivar and landrace accessions, while the cellulose CrI of foreign accessions was significantly higher than that of the Chinese accessions (Figure 1D). Interestingly, we found that there was a significant difference in cellulose CrI between the cultivars in China and those from other countries (Figure 1D). 

### 2.2. Genomic Variation, Population Structure, and LD (Linkage Disequilibrium)

A total of 326 wheat accessions, including 233 accessions from China,90 foreign accessions, and 3 accessions with unclear source, were used in this study (Figure 2A). The principal component analysis found that these accessions could be clustered into three clusters (Figure 2B). The admixture analysis also showed that there was also a population structure in this association panel (Figure 2C). Therefore, the principal components and kinship matrix were used in the following GWAS. A total of 2.09 M of high-quality single nucleotide polymorphisms (SNPs) were obtained from the association panel (Appendix A). The average marker density was 6.78 Kb/SNP, and the marker density ranged from 3.60 to 53.97 Kb/SNP in the different chromosomes (Appendix A). In detail, the chromosome chr6B had the highest marker density (3.60 Kb/SNP), while the chromosome chr6D had the smallest marker density. Furthermore, we found that the maker density in the D genome (37.10 Kb/SNP) was smaller than that in the A (6.85 KB/SNP) and B genomes (4.65 Kb/SNP), indicating that the diversity of the D genome is lower than that of the A and B genomes, which is in accordance with a previous study [24]. The LD decay in wheat is about 2.8 Mb (when r^2^ drops to half of the maximum value) (Figure 2D). The average marker density (340 Kb/SNP) is higher than the lowest number needed in this association panel. 

### 2.3. GWAS on Cellulose CrI

The fixed and random model circulating probability unification (FarmCPU) was used for cellulose CrIin the GWAS, and the suggested threshold was set at −log10(p) = 6, calculated with gec.jar software (https://pmglab.top/gec/#/, accessed on 1 October 2023). As a result, a total of 14 significant SNPs were found for cellulose CrI, and the SNPs were distributed at nine different chromosomes (1A(1), 2A(1), 2B(1), 2D(1), 3B(1), 3D(4), 4B(2), 5B(2), and 6A(1)) (Figure 3A,B). The significant SNPs were located mainly in the intergenic region (10) and upstream region (3) (Appendix A). The annotation of genes around the significant SNPs showed that *TraesCS3D03G0449100* encodes a serine/threonine-protein phosphatase PP1, while *TraesCS5B03G1126600* encodes probable pectinesterase. In addition, the significant SNPs could be integrated into 13 QTLs (Figure 3C). A total of 488 genes were annotated in the 13 QTLs, and the genes were unevenly distributed on the chromosomes (*qCrI1A.1*(65), *qCrI2A.1*(41), *qCrI2B.1*(48), *qCrI2D.1*(73), *qCrI3B.1*(6), *qCrI3D.1*(4), *qCrI3D.2*(5), *qCrI3D.3*(12), *qCrI3D.4*(30), *qCrI4B.1*(55), *qCrI5B.1*(53), *qCrI5B.2*(45), *qCrI6A.1*(51)). In addition, there were two consecutive significant SNPs in *qCrI4B.1,* and both of the significant SNPs were located upstream of *TraesCS4B03G0029800*. Moreover, *TraesCS5B03G1085500*, located at *qCrI5B.1*, encodes a stem-specific protein, TSJT1. Thus, *TraesCS4B03G0029800* and *TraesCS5B03G1085500* were selected as candidate genes. 

### 2.4. Haplotype Analysis of Candidate Genes

*TraesCS4B03G0029800*, annotated by two significant SNPs, encodes a two-pore K^+^ channel family protein (Figure 4A). Previous studies found that high amounts of K^+^ culm constituents were strongly associated with culm strength and lodging resistance, as K^+^ was found to be correlated with lignin deposition into the vascular bundles and sclerenchyma cells of the cell wall [4]. In detail, there are four SNPs in the promotor of *TraesCS4B03G0029800*, and these SNPs can be classified into four haplotypes (Figure 4B). H001 had 162 accessions, and 129 of them were cultivars, while 32 of them were landrace accessions. H002 had 41 accessions, and 35 of them were cultivars, while 6 of them were landrace accessions. H003 had purely landrace accessions, while H004 had only cultivar accessions (Figure 4C). Moreover, a comparison analysis was performed between the haplotypes. As a result, H002 showed significantly lower cellulose CrI, neutral detergent fiber (NDF) levels, and acid detergent fiber levels (ADF), while H003 showed no difference from H001 (Figure 4D).

The significant SNP chr5B:618051391 is located in the intergenic region between *TraesCS5B03G1085100* and *TraesCS5B03G1085500* (Figure 5A). *TraesCS5B03G1085500* was closer than *TraesCS5B03G1085100*. Moreover, *TraesCS5B03G1085500* encodes a stem-specific protein, TSJT1. So, *TraesCS5B03G1085500* was selected as the causal gene of this significant SNP. Two SNPs were found in *TraesCS5B03G1085500*, and they could be classified into four haplotypes (Figure 5B). In detail, H001 had more landrace accessions than H002 (Figure 5C). The comparison analysis between the different haplotypes showed that H002 had significantly lower cellulose CrI than H001, while H003 showed no difference with H001 (Figure 5D). 

In addition, there are 488 genes within LD (2.8 M) in 13 QTLs (Supplement Appendix A). The expression pattern of genes was downloaded at https://ipf.sustech.edu.cn/pub/plantrna/, accessed on 1 October 2023, and the tissue-specific gene expression (TAU) index was calculated [25] (Appendix A). As a result, 26 genes had a specific expression in the stems, and nine genes showed a highly specific expression in the stems (TAU value > 0.8) (Figure 6A,B). Among these nine genes, four showed a potential function and may influence cellulose CrI in wheat stems (Appendix A). In detail, *TraesCS6A03G0660000* encodes a WRKY transcription factor, and this WRKY transcription factor has been well-documented for the process of secondary cell wall formation [26]. *TraesCS5B03G1131000* encodes aspartic proteinase nepenthesin-1, and its protein has xylanase inhibitor N-terminal (PF14543) and xylanase inhibitor C-terminal (PF14541) domains. *TraesCS2B03G0225000* encodes a probable glucuronosyltransferase, and the homologous gene in rice is *OsIRX9*, which affects stem strength. The expression of *OsIRX9* in the *irx9* mutant resulted in xylosyltransferase (XylT) activity in the stems that was over double that of wild-type plants, and the stem strength of this line increased to 124% above that of the wild-type plants [27]. *TraesCS4B03G0034900* encodes a tubulin beta-8 chain, and the GUS results showed that *OsTUB8* is expressed in vascular bundles [28]. 

## 3. Discussion

Prior studies have noted that cell wall components are associated with wheat lodging, while cellulose crystallinity has not been widely characterized because of the difficulty in its measurement. This study first established a quick and high-throughput method for detecting cellulose crystallinity based on VIS/NIR spectra (Figure 7). Interestingly, we found that cellulose crystallinity showed a difference between accessions in China and other countries, and this difference was mainly caused by the cultivar accessions themselves. A possible explanation for the difference might be artificial selection because cellulose crystallinity is the major factor that negatively determines breaking-type lodging resistance [6]. This result could also be helpful for wheat lodging resistance breeding. In rice, many brittle culm mutants have been used for the genetic dissection of stem strength, and almost all the brittle culm mutants showed a significant decrease in cellulose crystallinity. In addition, the cloned genes of the brittle culm could be classified into four groups based on the cellulose. Group 1 (*OsBC6*, *OsBC7*) consists of the cellulose synthase that polymerizes the glucose into beta-1,4 glucan chains, group 2 (*OsBC3*) consists of the genes that modulate the transcription of cellulose synthase, group 3 (*OsBC14*, *OsBC25*) consists of the genes that affect the substrate of the cellulose synthase, and group 4 (*OsBC1*) consists of the genes that influence the function of cellulose synthase after translation. Meanwhile, no brittle culm gene has been reported in bread wheat. Hence, the genetic dissection of cellulose crystallinity is necessary for understanding stem strength in wheat.

Crystalline cellulose has many applications in many aspects, and its derivatives, such as microcrystalline cellulose and methylcellulose, play important roles in pharmaceutical, food, cosmetic, and other industries. For example, microcrystalline cellulose has been developed and is used in food ingredients [29]. Microcrystalline cellulose could be used for solid dosage formulations in drug delivery [30]. Furthermore, a composite generator film impregnated with cellulose nanocrystals could enhance triboelectric performance [31]. In addition, wheat straw is an important animal feed [32], and the crystalline structure of cellulose affects the rate of hydrolysis since its hemicellulose and its lignin make a bond with cellulose and therefore limit the process of hydrolysis [33]. A wheat stalk is a cheap and abundant lignocellulosic material, but most wheat straw is directly burned in cropland, which has caused serious atmospheric pollution in China [34]. It is of great significance to cultivate wheat with a reasonable cellulose crystallinity for wheat stem lodging resistance, and the biomass utilization of wheat straw and a balance between stem strength and biomass utilization could be an effective method. For example, the fragile culm 19 (FC19) mutation, with a significant decrease in cellulose CrI, largely improves plant lodging resistance and biomass saccharification in rice [35]. Overall, this rapid method for the measurement of cellulose crystallinity is the basis for improving stem structure, and the VIS/NIR model of cellulose CrI in this study will also contribute to the use of wheat stalks.

GWASs have been popularly used in dissecting complex traits in crops, and there are relatively few studies that have been reported on wheat because of its huge genome. In this study, a total of 2.09 M of high-quality SNPs were used for the GWAS with a high-throughput method for cellulose crystallinity measurements. In the present study, several QTLs were co-localized with previously identified QTLs. For example, *qCrI2D.1* and *qCrI4B.1* were co-located with *QTgw.crc-2D* [36] and *QTKW.ndsu.4B.1* [37], respectively, and *QTgw.crc-2D* and *QTKW.ndsu.4B.1* were detected with a grain weight in the thousands. Furthermore, *qCrI5B.2* and *qCrI6A.1* overlapped with *MU3_TEX_S4* (the third stage of the gray level in the image) and *THR_S2* (the ratio of the total projected area to the hull area), as detected by the same population as in our previous study [10]. In addition, the causal genes were hard to obtain in wheat because of the huge LD, but there are helpful methods for finding the most reliable genes. For example, *TraesCS2B03G0225000* has a specific expression in the stem, and its homologous gene in rice is *OsIRX9*, which is significantly associated with stem strength [27]. Thus, *TraesCS2B03G0225000* may affect stem strength as well. In other words, the combination of a GWAS, a high throughput phenotyping method, and multi-omics will help us understand and dissect traits.

## 4. Materials and Methods

### 4.1. Plant Materials 

The wheat association panel comprises 326 wheat accessions, mainly collected from China (230), America (5), and Australia (4). Detailed information on the 326 wheat accessions is described in Appendix A. The wheat association panel was planted in Wuhan, Hubei Province (30.39′ E, 114.30′ N), in November 2021 and harvested at the end of May the following year. All the materials were planted with two biological repeats, with two rows for each accession in each repeat. Field management followed local management guidelines. 

### 4.2. Spectrum Acquisition

Each whole plant was harvested, and then leaves, panicles, and roots were removed from the stalks. The stalks were then shaken and filtered with a 40-mesh sieve. The filtered powder was dried at 60 °C until it reached a constant weight. Finally, the power was used for VIS/NIR spectra collection with the Spectra Star 2600XT-R (https://www.kpmanalytics.com/brands/unity-scientific, accessed on 1 October 2023) at Huazhong Agricultural University. The wavelength of the spectra ranged from 680 nm to 2600 nm, with a resolution of 1 nm. Each sample was repeatedly loaded and scanned three times.

### 4.3. Measurement of Cellulose Crystallinity 

The powder was used for the measurement of cellulose crystallinity by an X-ray diffraction (XRD) method [15]. The powder was scanned under plateau conditions with a Rigaku-D/MAX instrument. The scan range and speed were 5–45° and 10°/min, respectively. The cellulose crystallinity was calculated according to the intensity values of the 110 peaks at 18.0° (*I_am_*) and the 200 peaks at 22.5° (*I*_200_) [38].
$$\mathrm{CrI}\left(\%\right)=\frac{I_{200}-I_{am}}{I_{200}}\times 100$$

### 4.4. Pretreatment of Spectra

ChemoSpec (version 6.1.9) was used to find the outliers of the raw spectra and perform the baseline correction [39]. Further, standard normal variation (SNV), multiple scatter correction (MSC), first derivative (FD), and Savitzky–Golay (SG) method were used for spectrum correction with R package pls (version 2.8-1) [40] and Prospectr (version 0.2.7) [41]. The Kenstone was used to split data into training and testing sets in an 8:2 ratio [41]. The support vector machine regression (SVR) model was used for the VIS/NIR model with R package e1071 (version 1.7) [42].

### 4.5. Genome-Wide Association Study

The genomic DNA of 326 core wheat lines was extracted, qualified, and sequenced on MGI 2000 using the standard protocol. Raw reads were processed for quality control and trimming with a FastQC tool (v0.11.7). The cleaning resulted in high-quality reads that were aligned to a wheat reference genome, IWGSC_ref 2.1 [43], with Burrows–Wheeler Aligner (BWA) software (https://bio-bwa.sourceforge.net/, accessed on 1 October 2023) [44]. The BAM alignment files were subsequently generated in samtools (https://github.com/samtools/samtools/releases/, accessed on 1 October 2023) [45]. Plink (1.9) was used to remove markers with minor allele frequencies (MAF) of <0.05 and a missingness per marker of >0.2. The principle components (PCs) were calculated by Plink (1.9) [46]. The linkage disequilibrium of the association panel was calculated with PopLDdecay software (3.41) [47]. The GWAS was performed using the FarmCPU model by rMVP [48]. The significant SNPs were annotated by SnpEff (5.0.1) [49]. Linkage disequilibrium block heatmaps were constructed using LDBlockShow software (1.40) [50]. gec.jar was used to find the threshold of the GWAS [51]. TBtools (1.120) was used to visualize the heatmap [52].

## 5. Conclusions 

In this study, we established an efficient method for the measurement of cellulose crystallinity based on VIS/NIR spectra, and this method will not only contribute to the quick measurement of cellulose crystallinity in wheat stems but will also benefit the application of the biomass saccharification of wheat stalks. Further, we performed a GWAS of a wheat association panel and identified candidate genes for cellulose crystallinity in wheat stems, which will contribute to a comprehensive understanding of stem strength. To summarize, this research provides a potential method to measure cellulose crystallinity and a genetic basis for lodging resistance breeding in wheat.

## Figures and Tables

**Figure 1 ijms-25-03028-f001:**
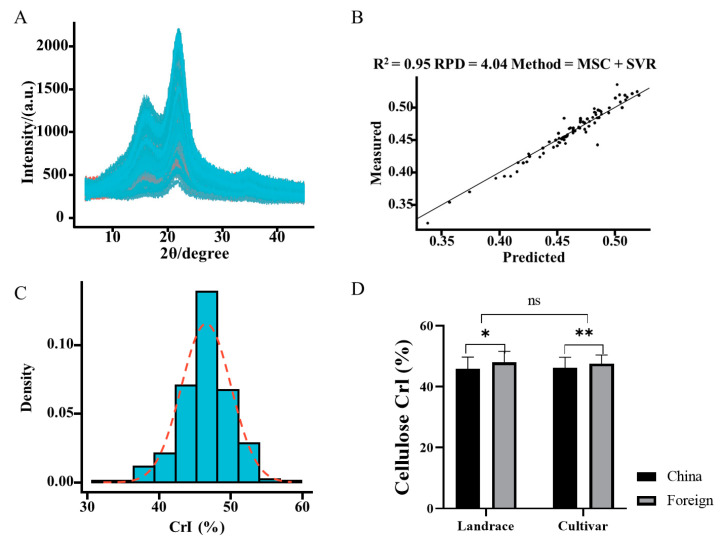
Distribution, prediction, and description analysis of cellulose CrI. (**A**) X-ray diffractograms; (**B**) the evaluation of the MSC pretreatment with the SVR model; (**C**) frequency distribution of cellulose CrI across the whole association panel; and the red dashed line is the fitted normal distribution curve; (**D**) comparison of cellulose CrI values among landrace and cultivar accessions in China and foreign countries. An unpaired student’s *t*-test was used in Figure 1D. * *p* < 0.05, ** *p* < 0.01.

**Figure 2 ijms-25-03028-f002:**
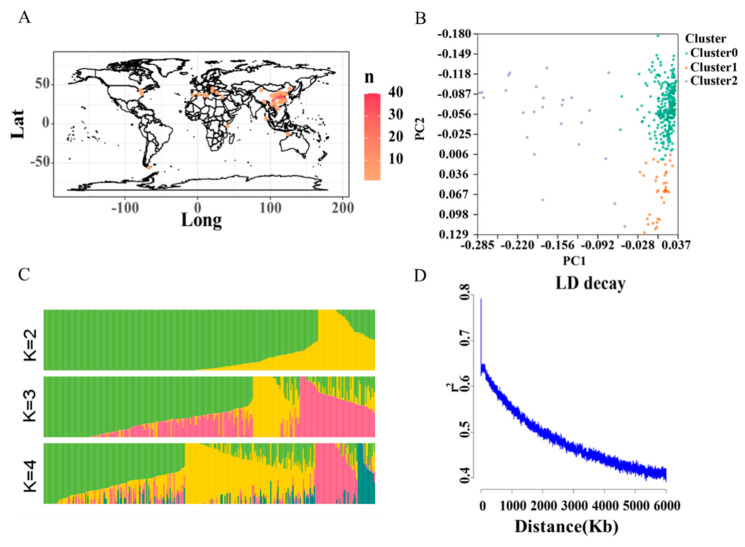
Distribution, principal component analysis, structure, and LD decay of the association panel. (**A**) Distribution of 326 wheat accessions; (**B**) principal component analysis; (**C**) structure analysis by Admixture software (https://dalexander.github.io/admixture/, accessed on 1 October 2023); and (**D**) LD decay analysis by PopLDdecay software (version = 3.41).

**Figure 3 ijms-25-03028-f003:**
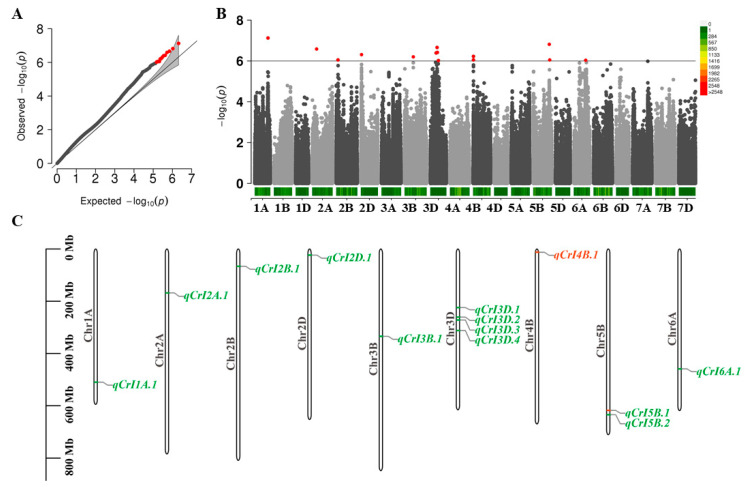
GWAS of cellulose CrI. (**A**) QQ plot of cellulose CrI; (**B**) Manhattan plot of cellulose CrI; and (**C**) distribution of 13 QTLs. The arrow represents the SNPs associated with the candidate genes, and the QTL in red had two significant SNPs.

**Figure 4 ijms-25-03028-f004:**
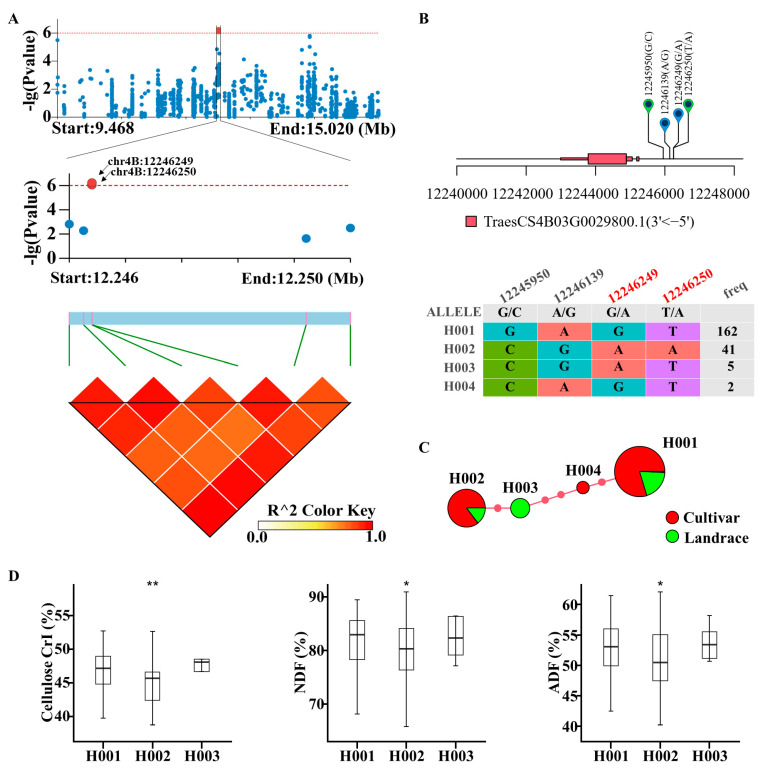
LD block and haplotype analysis of *TraesCS4B03G0029800.* (**A**) LD block of *TraesCS4B03G0029800* and (**B**) SNP information for each haplotype. (**C**) haplotype network of *TraesCS4B03G0029800* and (**D**) comparison of cellulose CrI, NDF, and ADF among *TraesCS4B03G0029800′*s haplotypes. NDF: neutral detergent fiber. ADF: acid detergent fiber. An unpaired student’s *t*-test was used in Figure 4D. * *p* < 0.05, ** *p* < 0.01.

**Figure 5 ijms-25-03028-f005:**
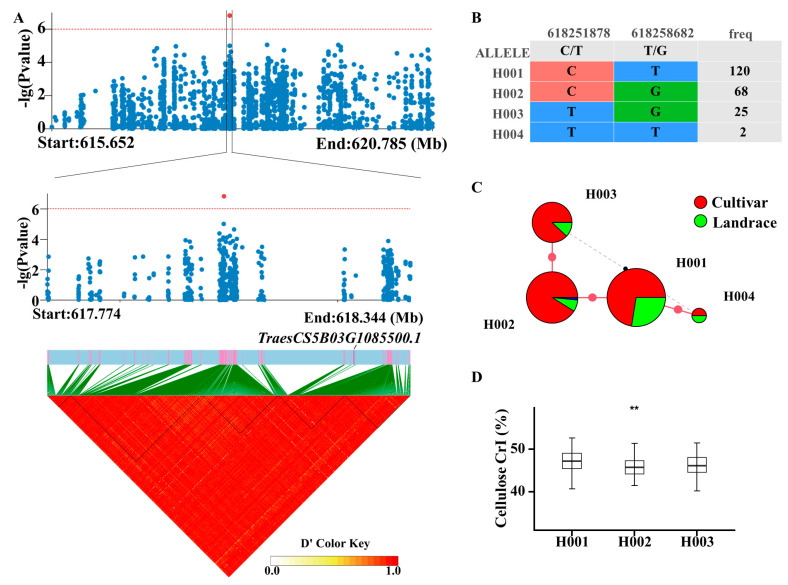
LD block and haplotype analysis of TraesCS5B03G1085500. (**A**) LD block of *TraesCS5B03G1085500* and the red dot represent the significant SNP. (**B**) SNP information for each haplotype. (**C**) Haplotype network of *TraesCS5B03G1085500*. (**D**) Comparison of cellulose CrI among *TraesCS5B03G1085500′*s haplotypes. An unpaired student’s *t*-test was used in Figure 5D. ** *p* < 0.01.

**Figure 6 ijms-25-03028-f006:**
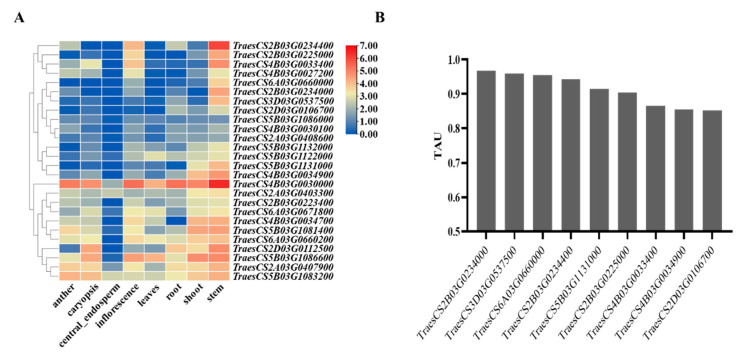
Expression pattern and TAU index of genes within 13 QTLs. (**A**) Heatmap of the expression pattern of 26 stem-specific expressed genes and (**B**) nine genes with a TAU index > 0.8.

**Figure 7 ijms-25-03028-f007:**
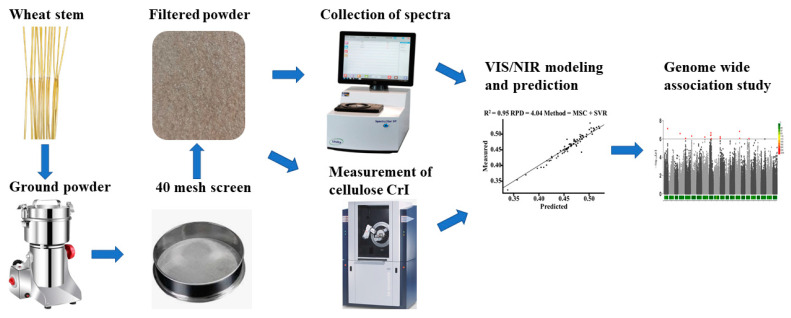
The pipeline of the sample collection, spectra collection, and data analysis.

## Data Availability

Additional data are provided as supporting information in the online version of this article.

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
