# Peer review of "Integrated VIS/NIR Spectrum and Genome-Wide Association Study for Genetic Dissection of Cellulose Crystallinity in Wheat Stems"

_ijms, 2024, doi:10.3390/ijms25053028_

Round 1

Reviewer 1 Report

Comments and Suggestions for Authors

The manuscript titled “Integrated VIS/NIR and GWAS for genetic dissection of cellulose crystallinity in wheat stem” by Li, J.; et al. is a scientific work where the authors employ a combination of the two techniques described in the title to discern the grade of crystalline of different wheat sources. The work can be interesting for a certain target audience albeit the main limitation that I found was related to the high-level concentration of analyzed wheat samples concentrated at one single region (China with 71.5 % of occurrence). The authors need to convice that this fact does not negatively impact on the data interpretation. The manuscript is generally well-written.

However, it exists some points that need to be addressed (please, see them below detailed point-by-point). The most relevant outcomes remarked by the authors can contribute in the growth of many fields like the development of more efficient materials to take part in many Industrial processes as indicated in the reviewing report. For this reason, I will recommend the present scientific manuscript for further publication in the International Journal of Molecular Sciences once all the below described suggestions will be properly fixed.

Here, there exists some points that must be covered in order to improve the scientific quality of the manuscript paper:

1) INTRODUCTION. “Wheat is one of the most important crops for the production of human food and lodging is a fundamental challenge in wheat cultivation” (lines 22-23). Some quantitative information should be furnished about the worldwide wheat production and its currently economic impact on society.

2)  “Recently, genome-wide association study (GWAS) has become a powerful tool (…) time-consuming” (lines 39-45). Here, the authors listed the potential capabilities of the cornestone technique used in this research. However, it may be desirable if its limitations were also remarked (as GWAS does not count the impact of environmental factors and the unability to identify all genetic determinants of complex traits, among other aspects).

3) RESULTS. Figure 1 (line 94) panel C. The gaussian fitting of the frequency distribution of the genetic dissection of cellulose crystallinity should be also plotted on the already depicted histogram. Then, the statistical studies carried out in the panels E and F should be also devoted in the data displayed in the panel D.

4) The authors should consider to add a table as Supplementary Information (SI) to correlate the gene expression levels to the grade of cristalline cellulose according to the wheat study population.

5) “A total of 2.09 M high-quality SNPs (…)” (line 108). The full-name (single-nucleotide polymorphisms) should be defined. Then, the abbreviation should be placed between brackets. Please, the authors should take this comment into account for the rest of the main manuscript body text.

6) DISCUSSION. The authors clearly states the interpration obtained of the achieved results. Nevertheless, in order to strengthen the significance of this work it may be desirable to list some potential applications of cristalline cellulose (and the subsequent neccesity to have a more complete outlook of those genetic factors which could modulate this cellulose nature) like excipient for drug delivery [1], sensing the intermolecular interactions exerted with other plant cell wall polymers [2], or the design of more durable composite materials with improved electrical performance [3].

[1] Viera-Herrera, C.; Santamaria-Aguirre, J.; Vizuete, K.; Debut, A.; Whitehead, D.C.; Alexis, F. Microcrystalline Cellulose Extracted from Native Plants as Excipient for Solid Dosage Formulations in Drug Delivery. Nanomaterials 2020, 1910, 10, 975. https://10.3390/nano10050975.

[2] Marcuello, C.; Foulon, L.; Chabbert, B.; Molinari, M.; Aguié-Béghin, V. Langmuir-Blodgett Procedure to Precisely Control the Coverage of Functionalized AFM Cantilevers for SMFS Measurements: Application with Cellulose Nanocrystals. Langmuir 2018, 34, 9376-9386. https://doi.org/10.1021/acs.langmuir.8b01892.

[3] Peng, J.; Zhang, H.; Zheng, Q.; Clemons, C.M.; Sabo, R.C.; Gong, S.; Ma, Z.; Turng, L-S. A composite generator film impregnated with cellulose nanocrystals for enhanced triboelectric performance. Nanoscale 2017, 9, 1428-1433.

7) MATERIALS AND METHODS. “The wheat association panel comprises 326 wheat accessions mainly collected from China (230), America (5), and Australia (4)” (lines 217-218). Almost all the gathered data comes from China. Did the authors consider that this fact could negatively affect to the reliability of the obtained results in this research? Some statements should be provided in this regard.

8) CONCLUSIONS. The main outcomes found in this research are clearly outlined in this section. The authors should consider to add some potential future action lines to pursue this research.

Comments on the Quality of English Language

The manuscript is generally well-written albeit it may be desirable if the authors could recheck it in order to polish final details susceptible to be improved.

Author Response

1) INTRODUCTION. “Wheat is one of the most important crops for the production of human food and lodging is a fundamental challenge in wheat cultivation” (lines 22-23). Some quantitative information should be furnished about the worldwide wheat production and its currently economic impact on society.

Thank you very much for your critical comments. We had added the worldwide wheat production information and its currently economic impact on society (line 22-24).

2) “Recently, genome-wide association study (GWAS) has become a powerful tool (…) time-consuming” (lines 39-45). Here, the authors listed the potential capabilities of the cornestone technique used in this research. However, it may be desirable if its limitations were also remarked (as GWAS does not count the impact of environmental factors and the unability to identify all genetic determinants of complex traits, among other aspects).

Thank you very much for your critical comments. We had added the limitations of GWAS (Line 43-46).

3) RESULTS. Figure 1 (line 94) panel C. The gaussian fitting of the frequency distribution of the genetic dissection of cellulose crystallinity should be also plotted on the already depicted histogram. Then, the statistical studies carried out in the panels E and F should be also devoted in the data displayed in the panel D.

Thank you very much for your critical comments and good suggestions. We had added the gaussian fitting of the frequency distribution of the genetic dissection of cellulose crystallinity and merged the figure 1D, 1E and 1F.

4) The authors should consider to add a table as Supplementary Information (SI) to correlate the gene expression levels to the grade of cristalline cellulose according to the wheat study population.

Thank you very much for your good idea. The expression levels could help in finding reliable genes, but we don’t have the gene expression data of the population. In the future study, we will sequence the expression data of the population and correlate the gene expression levels to the grade of crystalline cellulose.

5) “A total of 2.09 M high-quality SNPs (…)” (line 108). The full-name (single-nucleotide polymorphisms) should be defined. Then, the abbreviation should be placed between brackets. Please, the authors should take this comment into account for the rest of the main manuscript body text.

Thank you very much for your carefulness. We added the full-name (single-nucleotide polymorphisms) in line 114 and added other full-names of abbreviation such as line 86, line 92-96 .

6) DISCUSSION. The authors clearly states the interpration obtained of the achieved results. Nevertheless, in order to strengthen the significance of this work it may be desirable to list some potential applications of cristalline cellulose (and the subsequent neccesity to have a more complete outlook of those genetic factors which could modulate this cellulose nature) like excipient for drug delivery [1], sensing the intermolecular interactions exerted with other plant cell wall polymers [2], or the design of more durable composite materials with improved electrical performance [3].

Thank you very much for your suggestions. We added the potential applications of crystallinity cellulose and the genetic factors which could modulate the cellulose nature (Line 247-264).

7) MATERIALS AND METHODS. “The wheat association panel comprises 326 wheat accessions mainly collected from China (230), America (5), and Australia (4)” (lines 217-218). Almost all the gathered data comes from China. Did the authors consider that this fact could negatively affect to the reliability of the obtained results in this research? Some statements should be provided in this regard.

Thank you very much for your critical comments. In GWAS study, the composition of the association panel affects the results by influencing the population structure such as the indica and japonica subpopulation in rice. In GWAS studies, population structure is common and it could be removed by adding the covariance (kinship and principal components) into the model (Line 135).

Here are several examples.

[1] The association panel used in this study all collected from China.

Pang Y, Liu C, Wang D, et al. High-resolution genome-wide association study identifies genomic regions and candidate genes for important agronomic traits in wheat[J]. Molecular Plant, 2020, 13(9): 1311-1327.

[2] The association panel used in this study all collected from Japan.

Yano K, Yamamoto E, Aya K, et al. Genome-wide association study using whole-genome sequencing rapidly identifies new genes influencing agronomic traits in rice[J]. Nature genetics, 2016, 48(8): 927-934.

8) CONCLUSIONS. The main outcomes found in this research are clearly outlined in this section. The authors should consider to add some potential future action lines to pursue this research.

Thank you very much for your critical comments. We added several potential future action lines in the conclusions.

Reviewer 2 Report

Comments and Suggestions for Authors

L25 needs a reference

L44 needs a reference

L208 I think it is necessary to add (maybe in the supplementary materials) the names of the 326 accessions. Also add info about growth conditions, fertilizations and all the agronomic practices you followed during growing. I think this is important because it affects also cellulose content.

L 213 Please add the time point of harvesting

L215 Don't you think that the drying step can influence the spectra? Why you did not measure on fresh samples? Or direclty on plants without harvesting?

L235 I believe you have also to specify DNA extraction method and how you assessed quality and quantity of DNA.

Author Response

1) L25 needs a reference

Thank you very much for your critical comments. We added the reference.

Shah, L.; Yahya, M.; Shah, S. M. A.; Nadeem, M.; Ali, A.; Ali, A.; Wang, J.; Riaz, M. W.; Rehman, S.; Wu, W.; Khan, R. M.; Abbas, A.; Riaz, A.; Anis, G. B.; Si, H.; Jiang, H.; Ma, C., Improving Lodging Resistance: Using Wheat and Rice as Classical Exampl es. International journal of molecular sciences 20, (17), 4211.

2) L44 needs a reference

Thank you very much for your critical comments. We added the reference.

Zhang, W.; Yi, Z.; Huang, J.; Li, F.; Hao, B.; Li, M.; Hong, S.; Lv, Y.; Sun, W.; Ragauskas, A., Three lignocellulose features that distinctively affect biomass enzymatic digestibility under NaOH and H2SO4 pretreatments in Miscanthus. Bioresource technology 2013, 130, 30-37

3) L208 I think it is necessary to add (maybe in the supplementary materials) the names of the 326 accessions. Also add info about growth conditions, fertilizations and all the agronomic practices you followed during growing. I think this is important because it affects also cellulose content.

Thank you very much for your critical comments. The environment greatly affects the cellulose content. We added the names of the 326 accessions (supplement table S1) and the agronomic practices in line 290-291.

4) L213 Please add the time point of harvesting

Thank you very much for your critical comments. We add more specific time point of harversting in line 289.

5) L215 Don't you think that the drying step can influence the spectra? Why you did not measure on fresh samples? Or direclty on plants without harvesting?

Thank you very much for your critical comments. The drying step of the powder does influence the spectra because the water affects the absorption. The drying step could remove the effects of the water and make all samples at the same level.

As for the collection of spectra of fresh samples and directly on plants, these methods should also be feasible. But there are several restrictions. For one thing, the portable near-infrared spectrometer is uncommon. For another thing, the error of spectra of fresh samples and directly on plants is large. The spectra acquisition is affected by water, environment, humidity, sample fineness and so on. To collect the accurate spectra, the drying powder was also filtered with 40-mesh sieve in this study.

Here are some references that using the drying powder for spectra collection.

[1] Li X, Ma F, Liang C, et al. Precise high-throughput online near-infrared spectroscopy assay to determine key cell wall features associated with sugarcane bagasse digestibility[J]. Biotechnology for Biofuels, 2021, 14(1): 123.

[2] Zhang A, Hu Z, Hu X, et al. Large-scale screening of diverse barely lignocelluloses for simultaneously upgrading biomass enzymatic saccharification and plant lodging resistance coupled with near-infrared spectroscopic assay[J]. Industrial Crops and Products, 2023, 194: 116324.

6) L235 I believe you have also to specify DNA extraction method and how you assessed quality and quantity of DNA.

Thank you very much for your critical comments. We add the detailed information in line 316-321.

Reviewer 3 Report

Comments and Suggestions for Authors

A research article, "Integrated VIS/NIR and GWAS for genetic dissection of cellulose crystallinity in wheat stem," by Li et al., is well done in most parts. I believe it has some merit to future research. There are a few suggestions I would like to recommend.

line 39- it would be helpful if the authors could provide more explanation about the "Chinese spring reference genome" and perhaps include a link to the source.

Lines 55-56, "lodging...wheat" seems redundant and could be rephrased.

Figure 1F, it would be useful to explain what the different letters signify.

Figure 1D&F- More space between landrace and cultivar

Line 85- "p-value" should be capitalized consistently throughout the manuscript.

Line 126 and Figure 3B- The SNPs associated with the candidate genes can be highlighted in Figure 3B

Line 128- The threshold is based on? Bonferroni?

Line 162- remove bold

Conclusion- Future aspects and applications of the research can improve the conclusion based on the generated results.

Other thoughts- Other candidates with significant peaks in GWAS can be briefly introduced, and the genes can be listed in an Excel spreadsheet based on haplotype analysis, in addition to the two genes.

  Insert Retry    

Comments on the Quality of English Language

Just go over a few times

Author Response

1) line 39- it would be helpful if the authors could provide more explanation about the "Chinese spring reference genome" and perhaps include a link to the source.

Thank you very much for your critical comments. We add the explanation and link to the source in line 48.

2) Lines 55-56, "lodging...wheat" seems redundant and could be rephrased.

Thank you very much for your critical comments. We rephrase the sentence in line 65.

3) Figure 1F, it would be useful to explain what the different letters signify.

Thank you very much for your critical comments. According to other review comments, we merge the figure 1D, 1E, 1F and the letters was removed as well.

4) Figure 1D&F- More space between landrace and cultivar

Thank you very much for your critical comments. According to other review comments, we merge the figure 1D, 1E, 1F and we redraw the figure.

5) Line 85- "p-value" should be capitalized consistently throughout the manuscript.

Thank you very much for your critical comments. According to other review comments, we removed the p-value in line 85 and add the gaussian fitting line in the figure.

6) Line 126 and Figure 3B- The SNPs associated with the candidate genes can be highlighted in Figure 3B

Thank you very much for your critical comments. We highlighted the SNPs associated with the candidate genes in figure 3B.

7) Line 128- The threshold is based on? Bonferroni?

Thank you very much for your critical comments. In this study, the threshold is calculated by the gec.jar software (https://pmglab.top/gec/#/). The jec.jar could calculate the effective number of SNPs and provide suggested threshold. Compared to Bonferroni correction, it’s less strict and more effective.

Reference

Li, M.-X.; Yeung, J. M.; Cherny, S. S.; Sham, P. C., Evaluating the effective numbers of independent tests and significant p-value thresholds in commercial genotyping arrays and public imputation reference datasets. Human genetics 2012, 131, 747-756.

8) Line 162- remove bold

Thank you very much for your critical comments. We remove the bold letters here.

9) Conclusion- Future aspects and applications of the research can improve the conclusion based on the generated results.

Thank you very much for your critical comments. We add the future aspects and applications of the research in conclusion (line 333-338).

10) Other thoughts- Other candidates with significant peaks in GWAS can be briefly introduced, and the genes can be listed in an Excel spreadsheet based on haplotype analysis, in addition to the two genes.

Thank you very much for your critical comments. Other candidates with significant peaks were described in supplement table S5 and the related information of candidates were briefly introduced. In addition, the wheat genome is large and the SNP density in this study is 6.78 Kb/SNP which in not enough for haplotype analysis of all the genes, so we just choose the most promising genes. In the future, we will perform haplotype analysis of more candidate genes.

Round 2

Reviewer 1 Report

Comments and Suggestions for Authors

The authors did a great effort and the manuscript was improved. Nevertheless, the major concern about the data coming from one single region is still remaining.

Comments on the Quality of English Language

.